# Comprehensive Review of Functional and Nutraceutical Properties of *Craterellus cornucopioides* (L.) Pers.

**DOI:** 10.3390/nu16060831

**Published:** 2024-03-14

**Authors:** Mariana-Gabriela Bumbu, Mihaela Niculae, Irina Ielciu, Daniela Hanganu, Ilioara Oniga, Daniela Benedec, Mihaela-Ancuța Nechita, Vlad-Ionuț Nechita, Ioan Marcus

**Affiliations:** 1Department of Physiopathology, Faculty of Veterinary Medicine, University of Agricultural Sciences and Veterinary Medicine Cluj-Napoca, Calea Mănăştur 3-5, 400372 Cluj-Napoca, Romania; mariana-gabriela.bumbu@usamvcluj.ro (M.-G.B.); ioan.marcus@usamvcluj.ro (I.M.); 2Department of Infectious Diseases, Faculty of Veterinary Medicine, University of Agricultural Sciences and Veterinary Medicine Cluj-Napoca, Calea Mănăştur 3-5, 400372 Cluj-Napoca, Romania; mihaela.niculae@usamvcluj.ro; 3Department of Pharmaceutical Botany, Faculty of Pharmacy, “Iuliu Haţieganu” University of Medicine and Pharmacy, 23 Gheorghe Marinescu Street, 400337 Cluj-Napoca, Romania; 4Department of Pharmacognosy, Faculty of Pharmacy, “Iuliu Hațieganu” University of Medicine and Pharmacy, 12 Ion Creangă Street, 400010 Cluj-Napoca, Romania; ioniga@umfcluj.ro (I.O.); dbenedec@umfcluj.ro (D.B.);; 5Department of Medical Informatics and Biostatistics, Faculty of Medicine, “Iuliu Hațieganu” University of Medicine and Pharmacy, 6 Louis Pasteur Street, 400349 Cluj-Napoca, Romania; nechita.vlad@umfcluj.ro

**Keywords:** *Craterellus cornucopioides*, nutraceutical, phytochemical profile, functional activities

## Abstract

Nutraceuticals represent an emerging and dynamic scientific field due to their important potential in integrated healthcare through nutritional and medicinal approaches that interact and complement each other mutually. In an attempt to find new sources for such preparations, the present research focuses on the species *Craterellus cornucopioides* (L.) Pers. (Cantharellaceae), also known as the black trumpet. This wild mushroom species is renowned for its culinary excellence and unique taste and is used especially in a dehydrated state. However, beyond its gastronomic value, recent scientific investigations have revealed its potential as a source of bioactive compounds with pharmaceutical and therapeutic significance. Our study aimed, therefore, to review the current data regarding the morphology, chemical profile, and medicinal potential of the black trumpet mushroom, highlighting its unique attributes. By conducting a comprehensive literature analysis, this paper contributes to the broader understanding of this remarkable fungal species as a potential functional food and its promising applications in the field of therapeutics.

## 1. Introduction

Mushrooms or macrofungi are taxonomically categorized into two phyla within the subkingdom Dikarya: Basidiomycota (Agaricomycetes class) and Ascomycota (Pezizomycetes class) [1,2]. The diversity of taxonomically identified mushroom species is relatively limited, with approximately 7000 species cataloged, which represent around 10% of the overall existing number of fungal species [1,3]. Among these, a significant proportion, comprising 1–10%, are classified as toxic while the majority exhibit varying degrees of edibility, presenting either medicinal or nutritional interest and importance. Over 200 genera of macrofungi feature species are either used as food sources or appreciated for their potential health benefits [1,2,3]. These species can be found either cultivated or among the wild flora of different areas over the world, and their broad spectrum of high- and low-molecular-weight bioactive metabolites (belonging to classes of compounds such as the alkaloids, lipids, phenols, polysaccharides, proteins, peptides, steroids, lectins, and terpenoids) can be responsible for a large variety of therapeutic effects such as anti-inflammatory, antimicrobial, hepatoprotective, cytotoxic, antioxidant, antiviral, antifungal, and hypocholesterolemic effects [1,3].

The field of fungal biology plays a crucial part in advancing both the biotechnology and biomedical industries. Fungi assume a pivotal role in the realm of food technology, contributing substantially to the production of cultured food products through fermentation, exemplified by the likes of yeast, or through the direct consumption of specific fungal components. These organisms demonstrate an extraordinary level of metabolic precision, enabling the synthesis of a wide array of commodities, extending from food items to pharmaceuticals and vaccines, each having significant importance within the biotechnology sector. In parallel to the primary fungal products, the byproducts obtained during their processing, with progressive technological advancements driving heightened research attention towards these secondary outputs, are also of great importance. These byproducts undergo extraction and further processing, thereby affording their integration into functional foods or their use as active constituents within a diverse range of food products. The burgeoning consciousness surrounding health-related concerns has fueled a heightened interest in innovative health-promoting products, underlining the growing importance of fungal contributions within the medical and nutritional domains [4].

Mushrooms are valuable additions to the human diet. Among the widely consumed varieties are those belong to the genera *Agaricus*, *Auricularia*, *Dictyophora*, *Flammulina*, *Hericium*, *Lentinula*, *Pholiota*, *Pleurotus*, *Tremella,* and *Volvariella*. Notably, certain species belonging to the genera *Cantharellus*, *Sparassis*, *Lactarius*, *Suillus*, *Tuber,* and *Morchella* hold a special place for their culinary and health-related attributes [2]. The advantages of fungal species for dietary consumption surpass those of vegetables and are related to their high protein content, with all of them containing essential amino acids, high contents of chitin as a source of dietary fibers, high vitamin contents, and low lipid and cholesterol contents. Moreover, their nutritional value is enhanced by their tasty properties, physiological effects, and cultural characteristics [3,5,6,7,8].

On the other hand, mushrooms encompass various biologically significant active compounds with medicinal relevance, and various fungal species are acknowledged for their diverse medicinal attributes. Mushrooms with medicinal properties and therapeutic potential come from different ecological groups. These include xylotrophs like *Fomes fomentarius* (L.) J.J. Kickx., *Fomitopsis pinicola* (Sw.) P. Karst., *Ganoderma lucidum* (Curtis) P. Karst., *Grifola frondosa* (Dicks.) Gray, *Lentinula edodes* (Berk.) Pegler, *Pleurotus ostreatus* (Jacq.) Quél., *Trametes versicolor* (L.) C. G. Lloyd, and *Schizophyllum commune* (L.) Fr., as well as mycorrhiza-forming mushrooms like *Boletus edulis* Bull,, *Cantharellus cibarius* Fr., and *Tuber borchii* Vittad. [1,4,8].

*Craterellus cornucopioides* (L.) Pers., (Figure 1), a member of the Cantharellaceae family, was first named and described by Linnaeus in 1753 and is often known in English as the “black trumpet” or in French as the “trompette de la mort” [9]. Its common name derives from its trumpet-shaped morphology, boasting a tender flesh that spans a color spectrum from grey-black to brown. Its distinguishing features include an undulating rim and a hollow stem [10,11]. The mushroom cap has a narrow, wavy, and trumpet-shaped form; its hymenium is almost smooth and without folds; and its gills contain poorly defined folds, veins, or wrinkles while its stalk is narrow at the base, tubular, and pointed in shape [12]. It is a highly nutritious edible fungus that can be found worldwide [11,13,14]. This fungal species manifests its presence during the transitional period from summer to autumn, spanning from June to October. It predominantly thrives in acidic soil environments within deciduous forests characterized by oak and beech trees [15]. In its dehydrated state, the black trumpet is renowned for its culinary excellence, and its ground form yields a powder often termed the “truffle of modest means” [16]. It is known for its composition of phenolic compounds, sesquiterpenoids, β-glucans, polyunsaturated fatty acids, sterols, amino acids, minerals, organic acids, and stilbenes, which are responsible for its biological activities, among which the most well-known are the immunomodulating, anti-inflammatory, antioxidant, antimicrobial, and cytotoxic activities [8,11,13,17].

The continuous practices of the collection, cultivation, and commercialization of mushroom resources create numerous opportunities to study their nutraceutical and pharmacological potential for developing health-enhancing mycofood and myco-pharmaceuticals [1]. In this context, and considering the fact that in recent years, the species *C. cornucopioides* has been poorly studied despite its large accessibility and important nutritional and therapeutic potential [18], it appears important to resume its phytochemical composition and pharmacological activities. Therefore, the present paper reviews the current data regarding the morphology, chemical composition, and medicinal potential of this mushroom, highlighting its unique attributes. Thus, this review aims to contribute to the broader understanding of its value and potential applications in the medical and nutritional fields. By conducting a comprehensive analysis, we aim to contribute to our understanding of this remarkable species and shed light on its promising role in modern medicine and therapeutics. The novelty and originality of the present review lie in the fact that it reports the first study, to the best of our knowledge, that has gathered together the existing information on this edible fungal species, presenting nutritional and therapeutical potential. This research not only bridges the gap between traditional culinary practices and contemporary healthcare but also paves the way for future developments in the field of natural-product-based drug discovery.

## 2. Materials and Methods

A comprehensive review of the literature (4 October 2023) using the Preferred Reporting Items for Systematic Reviews and Meta-Analyses (PRISMA) criteria, including references published between 1978–2023, was conducted. Four databases, namely PubMed, Scopus, Google Scholar, and SpringerLink, were accessed. When searching in PubMed, a combination of keywords and the following MeSH terminology was used: (“Craterellus cornucopioides”) AND (“chemistry” [Subheading] OR “Pharmacologic Actions”[Mesh] OR “phytochemical properties” OR composition OR “chemical properties” OR compound* OR component* OR metabolite* OR bioactivity OR therapeutic OR “biological effect*” OR “pharmacological effect*” OR “pharmacological action*” OR pharmacology OR bioactive). For the Scopus, Google Scholar, and Spring-erLink searches, the following combination was used, searching within article titles, abstracts, or keywords: (craterellus AND cornucopioides).

In total, 180 references were found, of which 14 were on PubMed, 62 were on Scopus, 44 were on Google Scholar, and 60 were on SpringerLink. After removing duplicates (*n* = 29), article titles and abstracts were manually screened to exclude studies not related to the topic. A total of 73 manuscripts were eliminated after this preliminary screening. The remaining 78 full-text documents were thoroughly reviewed. The following inclusion criteria were used for selection: studies published in English, full-text availability, and the presence of the keyword Craterellus cornucopioides in the titles and/or abstracts. The exclusion criteria were as follows: the study topic (the absence of any information regarding the chemical composition or biological effects), conference papers, and records with no full text available. Finally, this assessment included 46 articles. The study selection process is displayed in Figure 2.

## 3. Results

### 3.1. Chemical Composition

The chemical compositions of different *C. cornucopioides* extracts and fresh vegetal mushroom is presented in Table 1. Metabolites from various classes of compounds can be found in different concentrations in extracts obtained from *C. cornucopioides* or in the fresh vegetal product. The vast majority of the studied and reported compounds are represented by vitamins, free amino acids, and minerals, all of these being valuable nutrients that justify the dietary use of this fungal species and, together with its functional activities, sustain its nutraceutical potential.

Phenolic compounds described in the composition of *C. cornucopioides* are included in different classes such as phenolic acids (e.g., ferulic, gallic, p-hydroxybenzoic, protocatechuic, chlorogenic, caffeic, syringic, and rosmarinic acids), flavonols (e.g., rutin, quercetin, myricetin), flavan-3-ols (e.g., catechin, epicatechin, epigallocatechin gallate) and stilbenes (e.g., t-resveratrol, piceatannol), and their concentrations vary in relation to the extract type, i.e., whether they are alcoholic (methanolic, ethanolic) or aqueous extracts [8,11,15,19,20,21,22,23].

Together with phenolic compounds, polysaccharides are the most cited compounds in the composition of *C. cornucopioides* aqueous extracts. CCPs-1 and CCPs-2 are two water-soluble acidic polysaccharides with different contents of uronic acid, rhamnose, fucose, arabinose, xylose, mannose, glucose, and galactose [13,14,18,24,25,26,27,28].

Other compound classes that are reported in ethyl acetate extracts of *C. cornucopioides* belong to the category of sesquiterpenoids, such as craterellins A–C; gymnomitr-3-en-10β,15-diol; illudins F, M, and T; and illudalenol [29,30]; but are also keto esters (e.g., 4-oxohex-1,6-diyl diacetate, 4-oxohex-5-enyl acetate, and 6-hydroxy-4-oxohexyl acetate) [31]. Furthermore, triacylglycerols (e.g., glycerol tri-dehydrocrepenynate) are found in the chloroforme:methanol extracts after removal of lipases with isopropanol [32].

As mentioned before, the nutritional value of *C. cornucopioides* is underlined by numerous studies that indicate vitamins, free amino acids, and minerals as the main compounds in the chemical composition of this species. One of the most important classes among these compounds is represented by vitamins, belonging to both of the traditionally categorized groups: water-soluble (e.g., vitamin C, vitamins of the B complex (B1, B2, B3, B6, B12)) and fat-soluble (e.g., vitamins A, D3, and E) vitamins. These essential, non-caloric, organic micronutrients were isolated from the *C. cornucopioides* methanolic, cyclohexane, or dichloromethane extracts [15,33]. A study performed by Watanabe et al. aimed to measure the vitamin B12 content in six different wild edible mushrooms commonly eaten by vegetarians from European countries. The *Lactibacillus delbrueckii*-based method indicated zero to trace levels of corrected vitamin B12 (0.01–0.09 µg/100 g dry weight, dw) in porcini mushrooms (*Boletus spp*.), parasol mushrooms (*Macrolepiota procera*), oyster mushrooms (*Pleurotus ostreatus*), and black morels (*Morchella conica*) while black trumpet (*C. cornucopioides*) and golden chanterelle (*Cantharellus cibarius*) mushrooms contained significant amounts of vitamin B12 (1.09–2.65 µg/100 g dw) [33]. Regarding fat-soluble vitamins, α-Tocopherol proved to be the most abundant fat-soluble vitamin in the cyclohexane extract of *C. cornucopioides* (2816 mg/100 g dry extract, corresponding to 47.90 mg/100 g dw of mushroom) [15]. Another study, performed by Liu et al. reports a significantly lower amount, 0.115 mg/100 g dw [21]. Group-D vitamins usually occur in two forms: ergocalciferol (vitamin D2, the major form of vitamin D found in edible mushrooms) and cholecalciferol (vitamin D3, found in lesser amounts). Significant amounts of vitamin D3 were found in the cyclohexane extract of *C. cornucopioides* (89.3 mg/100 g dry extract, which corresponded to 1.52 mg/100 g dw of mushroom) [15].

Along with vitamins, other classes of compounds with dietary importance such as free amino acids and minerals define *C. cornucopioides*’ chemical profile. The presence of both essential (lysine, threonine, methionine, valine, leucine/isoleucine, phenylalanine) and non-essential amino acids (arginine, serine, aspartic acid, glutamic acid, glycine, alanine, tyrosine, proline), except histidine and cystine, respectively has been reported in *C. cornucopioides* samples [15,21]. Glutamic acid is indicated as the major amino acid while variations in the total amino acid content and individual percentage are noticed when comparing literature data: 6.97 mg/g dw [15] and 67.48 mg/g dw [5]. Fruiting bodies of *C. cornucopioides* collected in Bosnia and Herzegovina contain a higher amount of non-essential amino acids (5.48 mg/g dw) compared to essential amino acids (1.49 mg/g dw), leading to an established ratio of 0.27, thus falling below the FAO/WHO-recommended value of 0.6 [13,15]. Moreover, the predominance of these amino acids is presented based on their associated tastes: bitter (arginine, iso-leucine, leucine, methionine, phenylalanine, and valine) (3.14 mg/g dw), followed by monosodium glutamate (MSG: glutamic and aspartic acids)-like (1.83 mg/g dw) and tasteless (lysine and tyrosine) (1.00 mg/g dw). The lowest contents have been found for the sweet-tasting amino acids (alanine, glycine, serine, and threonine) (0.68 mg/g dw) [15]. 

*C. cornucopioides* is also mentioned for containing high amounts of minerals such as Fe, Zn, K, Na, Ca, Mn, Cu, Mg, Pb, Cd, As, P, and Li [21,36,37,38,39]. A study performed by Dospatliev et al. on wild edible mushrooms grown in Bulgaria (*B. pinophilus*, *C. cibarius,* and *C. cornucopioides*) indicated N (36,517.33–37,923.20 mg/kg dw), K (30,555.6–31,454.93 mg/kg dw), P (2713.62–7347.93 mg/kg dw), and Mg (626.06–647.26 mg/kg dw) as the main essential elements in all tested samples belonging to the three fungal species. *C. cornucopioides* showed lower Fe, Zn, and Cu contents but also the highest Mn content compared to the other two mushrooms [34].

Furthermore, petroleum ether and chloroforme:methanol extracts of *C. cornucopioides* showed high amounts of sterols [21,34,35]. Ergosterol was the main sterol, appearing in almost double amounts when compared to *C. cibarius* (72.8 ± 0.4 vs. 42.4 ± 0.4% of total sterols). Other sterols proved to have lower amounts compared to *B. pinophilus* and *C. cibarius*, e.g., brassicasterol, campesterol, stigmasterol, and β-sitosterol. Cholesterol, on the other hand, proved to have the lowest percent of total sterols, 3.6 ± 0.1%, in *C. cornucopioides* extracts [34].

Not least, saturated fatty acids (SFAs), monounsaturated fatty acids (MUFAs), and polyunsaturated fatty acids (PUFAs) have been identified and quantified in large amounts in the cyclohexane extract of *C. cornucopioides*. The most abundant group of fatty acids comprised MUFAs (61.44%), followed by SFAs (24.08%) and PUFAs (14.48%) [15,21]. The main fatty acid in the tested mushroom was oleic acid [15], followed by stearic, palmitic, and linoleic acids [21,34]. Total fatty acids were represented by C 15:0 (pentadecanoic acid), C 16:0 (palmitic acid), C 16:1 (palmitoleic acid), C 17:0 (heptadecanoic acid), C 18:0 (stearic acid), C 18:1n9c (oleic acid), C 18:2n6c (linoleic acid), α-linolenic acid (C18:3n3c, ALA), C 20:0 (eicosanoic acid-arachidic acid), C 20:5n3 (eicosapentaenoic acid, EPA, omega-3 fatty acid), C 22:0 (behenic acid), C 22:6n3 (docosahexaenoic acid, DHA, omega-3 fatty acid), C 24:0 (lignoceric acid), C 24:1n15c, and important compounds of the SFA, MUFA, and PUFA groups [15,21,34]. Related to the presence of fatty acids, in particular of oleic acid and linoleic acid, *C. cornucopioides* could be recommended in treating hypertension and metabolic syndrome [15].

All these important classes of compounds are not only related to the nutritional profile but could also be relevant when exploring the pharmacological activities.

### 3.2. Functional Activities

*C. cornucopioides* is regarded as a valuable edible mushroom mostly for its nutritional characteristics and availability; still, several important functional activities were demonstrated for its extracts previously published studies (Table 2).

#### 3.2.1. Immunomodulatory Activity

A functional metabolite isolated from the *C. cornucopioides* aqueous extract, named CCP and represented by the fraction of polysaccharides, could serve as a natural immune modulator. CCP demonstrates potent immunomodulating properties, being capable of reversing immunosuppression by stimulating the development of immune organs and activating macrophages in the peritoneum. The immunoregulation induced by CCP operates through the TLR4-NF-κB pathway, leading to the increased mRNA expression of cytokines (IL-2, IL-6, TNF-α, and IFN-α). Furthermore, CCP partially activates spleen cells and mitigates spleen damage. Overall, this suggests that the administration of CCP has the potential to effectively enhance the immune competence of immunosuppressed individuals within the food nutrition industry [27].

Zhang et al. aimed to evaluate the protective effects of another fraction of polysaccharides, CCP2, on immunosuppressed mice, using a cyclophosphamide (Cy)-induced immunocompromised model. A total of 50 mice were included and randomly assigned to five groups (*n* = 10) through a double-blind experiment after a week of adaptive feeding. The groups comprised the normal control group, immunocompromised model group, and three CCP2 preventive-treatment groups at doses of 100, 200, and 400 mg/kg per day. On the 18th day of post-treatments, all mice were euthanized and their spleens and thymuses were dissected and weighed. The study design allowed for the comprehensive assessment of CCP2’s potential protective effects on immune function in the context of immunosuppression induced by cyclophosphamide. It has been reported that Toll-like receptors (TLRs) and NF-κBs play a crucial role in stimulating gene expression and cytokine secretion during immune responses. Considering the immunosuppressive effects of Cy-induced injury, the study aimed to uncover the underlying mechanism of CCP2’s impact via the TLR4-NF-κB-p65 signal pathways, known contributors to immune signaling cascades. Levels of TLR4, TRIF, and TRAF6 and the phosphorylation of P-NF-κB p65 were assessed. Following Cy administration, there was a significant decline in TLR4, TRIF, TRAF6, and P-NF-κB p65 levels compared to the NCG group. Upon CCP2 administration, these effects were alleviated in a dose-dependent manner, particularly at the 200 mg/kg per day dose. Furthermore, CCP2 treatment exhibited a concentration-dependent increase in the phosphorylation of p65. These results indicated that CCP2 activates the NF-κB signaling pathway, contributing to transcriptional activation [28].

The immunoregulatory effect and the underlying immunologic response mechanism of CCP with a triple-helix structure on peritoneal macrophages were also investigated in vitro. The study revealed that the treatment of peritoneal macrophages with 80 μg/mL CCP for 48 h significantly enhanced their phagocytic function and increased the activities of lysozyme (LZM), acid phosphatase (ACP), and succinodehydrogenase (SDH) in comparison to the untreated group. Western Blot and quantitative real-time polymerase chain reaction (qRT-PCR) assays demonstrated that 80 μg/mL CCP activated macrophages, leading to a significant increase in the mRNA expression of cytokines (IL-8, IL-1β, IFN-α, and TNF-α) and upregulation of the protein expression of the cell membrane receptor TLR4, along with its downstream protein kinase products (MyD88, TAK1, P-IKKα/β, and P-MEK) through the activation of the TLR4-NF-κB pathway in peritoneal macrophages. This suggests that the immunomodulatory mechanism of CCP in peritoneal macrophages is associated with the release of nitric oxide (NO) and related enzymes and cytokines by stimulating the NF-κB p50 pathway via TLR4-MyD88-TAK1 signaling [18].

#### 3.2.2. Anti-Inflammatory Activity

The anti-inflammatory effects of 27 mushroom extracts, obtained through either ethanol or hot water extraction, were assessed by measuring IL-6 production in lipopolysaccharide (LPS)-stimulated RAW264.7 cells. Six extracts (*C. croceus*, *R. mairei*, *C. tubaeformis*, *R. fellea*, *C. cornucopioides,* and *T. ustale*) prepared with ethanol showed a noteworthy decrease in IL-6 production and were chosen for further study. Specifically, ethanol extracts from five samples (*Russula mairei, Lactarius blennius, Craterellus tubaeformis, Russula fellea,* and *Craterellus cornucopioides*) significantly reduced IL-6 production at the 10 lg/mL concentration, ranging from 56.4% to 72.1% compared to LPS-stimulated control cells. Additionally, six wild mushroom ethanolic extracts (*R. mairei*, *L. blennius*, *C. tubaeformis*, *R. fellea*, *C. cornucopioides,* and *Trichloma ustale*) exhibited a dose-dependent decrease in nitric oxide (NO) production in LPS-stimulated RAW264.7 cells. At the 20 lg/mL concentration, NO production significantly decreased to between 44.0% and 66.7%. The anti-inflammatory effect, particularly that of *C. cornucopioides*, might be related to their pyrogallol content [40].

NLRP3 inflammasomes are groups of proteins in the cell’s cytoplasm that are mainly composed of ASC and caspase-1. ASC facilitates the activation of caspase-1, leading to the processing and release of pro-inflammatory cytokines like IL-1β and IL-18. The aim of a study performed by Xu et al. was to evaluate whether the polysaccharidic fraction CCPP-1 could activate NLRP3 inflammasomes. In the lipopolysaccharide (LPS) group, there were more NLRP3 and caspase-1 proteins. However, CCPP-1, especially at 400 μg/mL, reduced the production of activated caspase-1 and NLRP3. This suggests that CCPP-1 may control changes in inflammatory cytokine production by inhibiting NF-κB and NLRP3 signaling pathways [26].

The in vitro anti-inflammatory potential of different mushroom extracts was assessed through two methods: the inhibition of xanthine oxidase, involved in uric acid metabolism linked to conditions like gout, and determination of lipoxygenase activity inhibition, associated with the development of allergies and inflammatory processes. The extract order of xanthine oxidase inhibition activity was *T. melanosporum* > *C. cornucopioides* > *A. bisporus*, and for lipoxygenase inhibition, it was *C. cornucopioides* > *A. bisporus* > *T. melanosporum*. The inhibitory capacity increased with higher extract concentrations. This suggests the potential of these mushroom extracts to develop anti-inflammatory products to protect against neurological degeneration and control aging [22].

#### 3.2.3. Antimutagenic Effects

The ethanol extracts of *C. cornucopioides* were found to decrease the mutagenic activities of 2-Nitrofluorene (2-NF), Aflatoxin B1 (AFB1), Benzo(a)Pyrene (B(a)P), and 2-methoxy-6- chloro-9-(3-(2-chloroethyl)-aminopropyl)-aminoacridin (ICR-191) in a dose-dependent manner. The extract reduced the number of mutants induced by 2-NF without affecting survival, suggesting this effect was not due to killing bacteria. Interestingly, different doses of 2-NF were equally inhibited by a small dose of the extract, indicating a non-stoichiometric interaction. The antimutagenic effect decreased over time, suggesting that it does not interfere with ongoing mutations. The extract was classified as a desmutagen, inactivating or destroying mutagens, rather than a bio-antimutagen that interferes with mutation expression. Additionally, the *C. cornucopioides* extract completely inhibited the mutagenicity of AFB1, B(a)P, and ICR-191, which follow different metabolic pathways. The common polycyclic structure of the inhibited mutagens suggests a possible mechanism involving the formation of a complex between the desmutagen and the polycyclic mutagen. However, this extract could not inhibit the mutagenicity of Methyl methanesulfonate (MMS), N-Methyl-N’-nitro-N-nitrosoguanidine (MNNG), and 4-Nitroquinoline 1-oxide (4-NQO), suggesting different mechanisms for each mutagen. The desmutagenic factor appeared to be a common constituent in mushrooms, as other ethanol extracts from different mushrooms showed similar desmutagenic effects [41]. 

The ethanol extract of *C. cornucopioides* mushrooms exhibited potential antimutagenic effects against AFB1, (BaP), acridine half-mustard ICR-191, and 2-NF. This effect is likely attributed to a direct chemical interaction with the mutagen or the inhibition of the activation process of promutagens [42].

**Table 2 nutrients-16-00831-t002:** The functional activities of *C. cornucopioides* extracts or vegetal products.

Functional Activity	Type of Study	Mechanism (s)	Extracts or Tested Vegetal Products	Reference
Antimicrobial	In vitro	Biofilm inhibition	Extracts	[43]
	In vitro	Inhibitory activity against reference strains of *Staphylococcus aureus, B. subtilis, B. cereus, Escherichia coli, Proteus mirabilis, Aspergillus niger, Candida albicans, Penicillium italicum, Mucor mucedo, Trichoderma viride*	Acetone extract	[8]
	In vitro	Antibacterial activity on all tested microorganisms—reference strains of *B. subtilis, E. faecalis, B. licheniformis, A. tumefaciens, E. coli, S. aureus*	Water and methanol extracts	[19]
Anti-inflammatory				
Immunomodulatory	In vitro	RAW264.7 cell activation with enhanced phagocytosis	Isolated polysaccharides	[28]
	In vivo mice models	Enhanced immunoregulatory activity Significantly increased spleen and thymus weight indices of the BALB/c mice models Synergistic effects on the T- or B-lymphocyte proliferation induced by ConA or LPS, respectively Enhanced natural killer (NK) cell activity Significantly increased phagocytic activity of peritoneal macrophages in immunosuppressive mice	Isolated polysaccharides	[27]
	In vivo	Protective function against immunosuppression induced by cyclophosphamide Immunoregulatory activity in immunosuppression BALB/c mice model Macrophage activation via enhanced production of cytokines (IL-2, IL-6, and IL-8)	Isolated polysaccharides	[28]
Antioxidant	In vivo simulated digestion model	Reduced DPPH-radical-scavenging activity, reducing power, and metal chelating activity due to the processing methods	Processed products (steaming, boiling, frying, and microwaving)	[17]
		Free radical scavenging, superoxide anion scavenging, and reducing power	Acetone extract	[8]
	In vitro	Strong scavenging abilities on DPPH and ABTS radicals. Oxidative hemolysis induced by AAPH in mice erythrocytes was effectively reversed by incubation with CCPP-1 Protective effect against AAPH-induced oxidative stress in erythrocytes	Polysaccharide fraction (CCPP-1)	[13]
	In vitro	Free-radical-scavenging effect	Extracts	[43]
	In vitro	DPPH-radical-scavenging activities enhanced with elevated concentrations (methanolic extracts > water extracts)	Water and methanol extracts	[15,19]
Cytotoxic	In vitro	Against MCF-7 breast cancer cell line	Extracts	[43]
	In vitro	Moderate cytotoxic effect on cancer cell lines (human epithelial carcinoma HeLa, human lung carcinoma A549, human colon carcinoma LS174); HeLa cell was most sensitive No cytotoxic effect on normal cells	Acetone extract	[8]
	In vitro	Some degree of cytotoxicity over HepG2 cell line Cytotoxic effects of the extracts increased with elevated concentrations Methanolic extracts had the lowest IC50 values, and thus, methanolic extracts > water extracts	Water and methanol extracts	[19]
	In vitro	Cyclohexane and dichloromethane extracts expressed significant cytotoxic activity against human epithelial cervical cancer cells (HeLa), adenocarcinomic human alveolar basal epithelial cells (A549), colorectal cancer cells (LS174), and normal MRC-5 human embryonic lung fibroblast cells	Cyclohexane and dichloromethane extracts	[15]
Angiotensin-converting enzyme (ACE)-inhibitory	In vitro	Strong and dose-dependent ACE-inhibitory activity found only for the aqueous extract (IC50 = 0.74 μg/mL)	Aqueous and methanol extracts	[15]

#### 3.2.4. Antioxidant Activity

In a study performed by Palacios et al., *C. cibarius* and *C. cornucopioides* exhibited the most significant antioxidant capacity. Analyzing the phenolic compounds found in the composition of the studied mushrooms, *C. cornucopioides* contained the highest level of myricetin while *C. cibarius* showed higher amounts of caffeic acid and catechin compared to other species. This observation suggests that these specific phenolic compounds might possess stronger antioxidative capabilities than others found in these mushrooms. Even in relatively small quantities, they appeared to inhibit linoleic acid oxidation more effectively than compounds found in higher amounts [20].

The evaluation of antioxidant activity involved examining free radical scavenging, superoxide anion radical scavenging, and reducing power. The DPPH radical was used to gauge the free-radical-scavenging activity in lichen extracts. Several studies have reported antioxidant activity for *C. cornucopioides*, using different extraction solvents. In the study performed by Kosanić et al., the antioxidant activity of selected mushrooms was affirmed through acetone extraction. Significant amounts of polyphenols found in *C. cornucopioides*, such as gallic acid, quercetin, rutin, catechin, and p-coumaric acid, suggest their potential role in the potent antioxidant activity [8].

Two polysaccharidic fractions (CCPs-1 and CCPs-2) were isolated and purified from *C. cornucopioides*, with chemical composition analysis revealing distinct molar ratios of rhamnose, fucose, arabinose, xylose, mannose, glucose, and galactose in both polysaccharidic fractions. Structural elucidation through Fourier-Transform Infrared Spectroscopy (FT-IR), periodate oxidation, Smith degradation, methylation analysis, and Nuclear Magnetic Resonance (NMR) indicated a primary connection by mannose with (1→3)- linked configuration. The configuration of the two polysaccharides demonstrated a random coil with a pyranoid polysaccharide containing α or β glycosidic bonds. To enhance antioxidant activity, carboxymethylated polysaccharides (cmCCPs-1 and cmCCPs-2) were obtained, featuring degree-of-substitution values of 0.34 and 0.52. Structural analysis revealed nonselective carboxymethylation, with partial substitution at C-2, C-4, or C-6. Remarkably, cmCCPs-2 exhibited the most significant antioxidant activity [24].

Another polysaccharide fraction, named CCPP-1, isolated from *C. cornucopioides* primarily consisted of mannose, glucose, xylose, arabinose, and fructose in a molar ratio of 0.7:0.05:0.18:1:0.05. In vitro antioxidant activity assays revealed that CCPP-1 exhibited robust scavenging capabilities against DPPH and ABTS radicals. When exposed to AAPH-induced oxidative hemolysis in mice erythrocytes, CCPP-1 effectively reversed the damage. Additionally, CCPP-1 significantly mitigated AAPH-induced intracellular reactive oxygen species (ROS) generation. Furthermore, CCPP-1 played a significant role in restoring the AAPH-induced elevation of intracellular antioxidant enzyme activities, such as those of glutathione peroxidase (GPx) and catalase (CAT), to normal levels. It also inhibited the formation of intracellular malondialdehyde (MDA). Therefore, CCPP-1 demonstrated protective effects against AAPH-induced oxidative stress in erythrocytes, suggesting its potential as a naturally derived antioxidant agent [13].

#### 3.2.5. Cytotoxic Activity

Research performed by Radović et al. delved into the chemical composition of the *C. cornucopioides* mushroom, examining vitamins, fatty acids, 5′-nucleotides, nucleosides, amino acids, and in vitro biological activities like the antioxidant and angiotensin-converting enzyme (ACE)-inhibitory and cytotoxic activities. *C. cornucopioides* was found to have low energy, fat, and carbohydrate contents, but it was rich in dietary fibers, especially β-glucan, niacin, and α-tocopherol. The essential and non-essential free amino acid contents were 1.49 and 5.48 mg/g dw, respectively while nucleosides and 5′-nucleotides were determined at 1.84 and 3.99 mg/g dw. Unsaturated fatty acids (UFAs) made up 75.92%, with oleic acid being the major UFA. Significant cytotoxic activity was observed in cyclohexane and dichloromethane extracts against various cell lines. The aqueous extract demonstrated strong ACE-inhibitory activity, with an IC50 of 0.74 μg/mL. It was therefore proven that the *C. cornucopioides* mushroom represents a valuable nutrient source comprising vitamins, dietary fibers, amino acids, nucleotides, and fatty acids, contributing to its overall nutritional profile, indicating potential for ACE-inhibitory activity and suggesting a possible application in anti-hypertensive diets [15].

Another study conducted by Kol et al. explored both the bioactive components of this mushroom species, including phenolics, flavonoids, β-carotene, and lycopene, as well as their ability to scavenge DPPH radicals. Additionally, the study examined the impact of the mushroom on inhibiting cell growth in HepG2 (hepatocellular carcinoma) cells and its antibacterial effects. The findings revealed that methanol extracts contained a higher phenolic content (37.71 ± 1.42 μg/mg) compared to water extracts (13.78 ± 1.60 μg/mg). Methanolic extracts also exhibited greater levels of β-carotene and lycopene, along with higher DPPH-scavenging activity (IC50: 5.26 ± 0.67 mg/mL). Conversely, water extracts had a higher flavonoid content (2.13 ± 0.06 μg/mg). *C. cornucopioides* demonstrated significant cell-growth-inhibitory effects on HepG2 cells, with an IC50 of 18.41 ± 1.10 mg/mL for aqueous extracts and 3.14 ± 1.07 mg/mL for methanolic extracts [19].

The examined dry extracts of the *C. cornucopioides* mushroom served as a rich reservoir of bioactive compounds with antioxidant properties. Notably, the aqueous dry extract exhibited the highest total phenolic content and superior antioxidant activity, as assessed through DPPH, ABTS•+, and ferric reducing-power assays. Alcoholic dry extracts are significant sources of free sterols. The cytotoxic activity against *Daphnia magna* crustaceans followed a decreasing order: ethanolic extract ~ methanolic extract > aqueous extract. All analyzed dry extracts manifested a selective cytotoxic effect, particularly targeting human epidermoid A431 tumor cells [11].

Human cervical cancer cells (HeLa), human lung carcinoma (A549) cells, and human colon carcinoma (LS174) cells were used to assess the cytotoxic activity of *C. cornucopioides.* The impact on cancer cell survival was assessed 72 h after the extract addition using an MTT assay. The mushroom extract displayed a moderate cytotoxic effect on the cancer cells used, with HeLa cells showing the highest sensitivity. There is limited information available on the anticancer potential of the *C. cornucopioides* species. No cytotoxic effect of *C. cornucopioides* on normal cells was detected, emphasizing the selective nature of the mushroom’s cytotoxic potential [8].

Methanolic extracts of 29 distinct wild edible mushrooms were investigated for their antioxidant, antiproliferative, cytotoxic, and pro-apoptotic effects on the lung adenocarcinoma cell line A549. Specific species displayed notable antioxidant activity correlated with their elevated total phenolic content. Among them, *C. cibarius*, *Cantharellus cinereus*, *Craterellus cornucopioides,* and *Hydnum repandum* (order Cantharellales) exhibited significant cytotoxicity and induced apoptosis in A549 cells [1]. Among the wild mushrooms evaluated in another study performed by Vasdekis et al., *Cantharellus cibarius* and *Craterellus cornucopioides* displayed the highest cytotoxicity against the A549 cell line. The cytotoxicity of authentic piceatannol (a stilbenic compound, resveratrol metabolite) was also assessed against the A549 cell line, showing a significant reduction in cell viability even at extremely low concentrations [23].

Five human cancer cell lines—SK-BR-3 (breast cancer), SMMC-7721 (hepatocellular carcinoma), HL-60 (human myeloid leukemia), PANC-1 (pancreatic cancer, and A-549 (lung cancer)—were subjected to analyses of the cytotoxicity of *C. cornucopioides* cultures for 48 h. Each test was conducted twice, using cisplatin as a positive control. From the cultures of the species, three illudin sesquiterpenoids (craterellins A–C) and one gymnomitrane sesquiterpenoid (gymnomitr-3-en-10β,15-diol) were isolated. Additionally, four previously reported compounds—illudin F, illudin M, illudin T, and illudalenol—were identified. The structures of the new compounds were elucidated through extensive spectroscopic analysis. The cytotoxic activities of these compounds were assessed on the five tumor cell lines, revealing that craterellin C demonstrated moderate cytotoxicity against A-549 with an IC_50_ value of 21.0 μM [29]. On the other hand, cultures of *C. cornucopioides* yielded two new illudane sesquiterpenoids, craterellins D and E, along with a new menthane monoterpene, 4-hydroxy-4-isopropenyl-cyclohexanemethanol acetate. Compounds were subjected to cytotoxicity assessments against five human cancer cell lines (HL-60, SMMC-7721, A-549, MCF-7, and SW-480) using the MTT method. Unfortunately, none of the compounds exhibited significant activity at a concentration of 40 μM [30].

#### 3.2.6. Antimicrobial Activity

Regarding the examination of antibacterial properties, it was proven that extracts from *C. cornucopioides* display significant antimicrobial capacities and demonstrate antibacterial effects against microorganisms such as *B. subtilis*, *E. faecalis, B. licheniformis, A. tumefaciens*, *E. coli,* and *S. aureus*. Both water and methanolic extracts of *C. cornucopioides* inhibited the growth of microorganisms, with inhibitory zone (IZ) values ranging from 6 to 8 mm in length [19].

The extract of *C. cornucopioides* showcased its efficacy against a large range of microorganisms, demonstrating a minimum inhibitory concentration (MIC) ranging from 0.1 to 0.2 mg/mL for bacteria and 5 to 10 mg/mL for fungi. Among the tested microorganisms, *Bacillus cereus* and *Bacillus subtilis* proved to be the most sensitive, exhibiting an MIC value of 0.1 mg/mL. The antimicrobial impact of *C. cornucopioides* depended on both the concentration of the extract and the specific microorganisms under examination. The experiments unveiled antibacterial activity against both Gram-positive and Gram-negative bacteria, with the latter showing greater resilience—a consistent pattern in microbiological studies where Gram-negative bacteria often exhibited increased resistance. Fungi, characterized by a more intricate cell wall structure, displayed heightened resistance compared to bacteria. These observations align with existing literature, corroborating the antimicrobial potential of *C. cornucopioides*. The mushroom’s composition, including quercetin, ferulic, gallic, and p-coumaric acids, is recognized for potent antimicrobial properties against a diverse array of bacteria and fungi, reinforcing the anticipated effectiveness of *C. cornucopioides* in microbial inhibition [8].

## 4. Discussion

*C. cornucopioides* is an edible fungal species with remarkable functional properties. The metabolites that are responsible for these functional properties are polysaccharides, phenolic compounds, terpenes, amino acids, fatty acids, sterols, and proteins (Table 1), which exhibit antioxidant, antimicrobial, anticancer, anti-inflammatory, anti-hypertensive properties (Table 2) [9,12]. Moreover, the high nutritional content of the mushroom and its favorable health characteristics make it an important nutraceutical [12]. The functional importance of this fungal species is highly correlated to its chemical composition (Figure 3). Due to the lack of knowledge regarding the cultivation, cooking methods, harvesting, storage, and processing of this fungal species, it is not receiving the interest it deserves [9]. The present review offers an overview of its chemical composition and functional activities, providing the necessary arguments for more detailed studies that may emphasize its importance.

Compared to other species of the genus, *C. cornucopioides* is richer in polyphenolic compounds. Butkhup et al. showed that the myricetin amount of *C. aureus* is significantly lower than that of *C. cornucopioides* (0.03 ± 0.0 vs. 35.91± 0.98) [6]. *C. cornucopioides* has also been proven to have higher amounts of total polyphenols and flavonoids, correlated with a significantly higher antioxidant activity in the DPPH assay, compared to *C. aureus* [6,8]. Limited studies regarding the individual profiles of phenolic compounds in wild edible mushrooms have been carried out as cultivated mushroom species are known to be better represented in terms of content. As numerous wild edible mushrooms are scarcely investigated and the existing studies have proven important amounts of phenolic compounds, it appears that *C. cornucopioides* may be considered a fungal species with important potential for antioxidant capacity [6,8,20].

When compared to other fungal species such as *Agaricus campestris*, *Boletus edulis*, *Calocybe gambosa*, *Cantharellus cibarius*, *Entoloma clypeatum*, *Flammulina velutipes*, *Macroleptiota procera*, *Morchella elata,* and *Pleurotus ostreatus*, it appears that *C. cornucopioides* is richer in proteins than all these species and contains significant amounts of fat and total sugars. At the same time, the edible portion of *C. cornucopioides* provides the highest energy among these species when compared to *Clitocybe maxima*, *Catathelasma ventricosum*, *Stropharia rugoso-annulata,* and *Laccaria amethystea* [5,21]. The protein content of the species is also significantly higher when compared to *Boletus pinophilus* and *Cantharellus cibarius* [34].

In previous studies, *C. cornucopioides* proved to contain important amounts of N, K, Na, Ca, Ni, Mn, and Co among essential elements [34,44]. Compared to *Boletus pinophilus* and *Cantharellus cibarius*, *C. cornucopioides* revealed higher amounts of N, K, Na, Ca, Ni, Mn, and Co [34]. The high amounts of these minerals are directly related to efficient metabolic reactions, the transmission of impulses, bone formation, and the regulation of the water-and-salt balance [44]. The macro- and microelements have been highly investigated via comparison with several mushroom species highly known for their nutritional value, such as *Agaricus spp., Boletus edulis, Pleurotus ostreatus, Cantharellus cibarius,* and *Lentinus edodes* [37,38,39].

Vitamin B12 is the vitamin compound that is found to be in high amounts in *C. cornucopioides* compared to *Boletus* spp., *Macrolepiota procera*, *Pleurotus ostreatus,* and *Morchella conica*. It is reported that *C. cornucopioides* contains the authentic B12, but not the pseudo B12 that is inactive in humans, in this way proving that *C. cornucopioides* could be an important and useful source of the B12 vitamin, which is essential especially for vegetarians. A moderate intake of this mushroom may prove, in this way, to be beneficial for the prevention of severe B12 deficiency in vegetarians [33].

The nutritional value of the species may also be sustained by its high total lipid content compared to *Agaricus arvensis*, *Cantharellus cibarius*, *C. lutescens*, *Hericium erinaceus*, *Hydnum repandum*, *Lactarius deliciosus*, *Pleurotus pulmonarius,* and *Ramaria flavescens.* This allowed to calculate an average value of around 400 kcal/100 g dried mushroom, allowing to rebalance or supplement menus rich in lipids or to integrate this mushroom in hypocaloric diets [39].

The difference between the identified and quantified quantities of monounsaturated fatty acids (MUFAs) and polyunsaturated fatty acids (PUFAs) is considered to be a positive ratio as SFAs are associated with cardiovascular disorders and atherosclerosis while the high amounts of PUFAs and MUFAs are related to the prevention of these disorders. The main fatty acid reported was oleic acid, a monounsaturated non-essential fatty acid effective in lowering cholesterol levels. Essential fatty acids were represented by linoleic acid (C18:2n6c) and α-linolenic acid (C18:3n3c), which cannot be synthesized by the human organism and must be provided by dietary sources. Linoleic acid was found, in *C. cornucopioides,* to have a relative content of 10.85%. Stearic acid, on the other hand, known for the lack of effect in terms of increasing serum low-density lipoprotein (LDL) cholesterol levels, was the most abundant (12.43%) among the identified saturated fatty acids. Nevertheless, a higher content of unsaturated than saturated fatty acids can sustain the fact that *C. cornucopioides* can be classified as a healthy food, as low-calorie and low-fat diets are recommended in cases of high blood cholesterol levels and dietary sources with high linoleic and oleic acid levels are known to help in preventing atherosclerosis [15,21,34,39].

On the other hand, it appears that this species is among the species with the lowest content of selenium compared to species of the *Boletus* genus or varieties of *Agaricus bisporus* [45]. Ergosterol is also found in lower amounts compared to *Agaricus bisporus*, *Hygrophorus marzuolus*, *Pleurotus ostreatus*, *Calocybe gambosa,* and *Lentinus edodes*. Regarding the biological activities, which are correlated to the chemical composition, *C. cornucopioides* holds a special role, proving a selective anti-inflammatory activity by decreasing the production of NO and IL-6 but not TNF-a in LPS-stimulated RAW264.7 cells, together with *Russula mairei*, *Lactarius blennius*, *Craterellus tubaeformis,* and *Russula fellea* among 27 other tested mushrooms [40].

Unanimously described as an excellent source of dietary fibers and proteins, as well as of nutrients such as vitamins B1, B2, B12, C, D, and E; niacin; folate; and minerals while being low in fat and calories, *C. cornucopioides* manifests important functional properties. In this regard, immunomodulatory, antioxidant, anti-inflammatory, antimutagenic, cytotoxic, antimicrobial, and ACE-inhibitory activities are reported for *C. cornucopioides*-derived extracts. All these bioactivities deserve comprehensive assessment for a better understanding of the underlying mechanisms of action and related bioactive compounds. The available literature is relatively limited; still, the immunomodulating capacity has been highlighted by both in vitro and in vivo results [18,27,28]. This was the case for the two polysaccharide fractions isolated from a *C. cornucopioides* aqueous extract that were suggested as potent natural immune modulators able to reverse induced immunosuppression and activate cell-mediated immunity [27]. In addition, anti-inflammatory and broad-spectrum antimicrobial and antifungal properties were demonstrated in vitro for derived extracts. Another relevant potential refers to the inhibitory effect on tumoral cells by acetone [8], methanol [15,19], cyclohexane, and dichloromethane extracts [15].

The most important functional activities of the species, the antioxidant, anti-inflammatory, anti-hyperglycemic, and the cytotoxic activities, are strongly correlated with different classes of metabolites such as the flavonoids and phenolic compounds, fatty acids, sterols, and minerals. All these compounds are highly related and sustain the nutritional importance of the species [5,46]. In order to support the unique value of *C. cornucopioides*, all correlations are emphasized via comparison with several mushroom species highly known for their nutritional value (Table 3). It can be clearly observed, by analyzing the few existing studies that have realized these correlations in comparison with other edible mushrooms [20,21,22,23], that *C. cornucopioides* is either the most active or is situated amongst the most active and metabolite-rich edible mushrooms, highlighting, once more, its great importance as a functional food.

The limitations of the present studies are represented by the inconsistencies of the scientific literature. First of all, a limited number of studies treating the fungal species exist, and this reduces, significantly, the amount of evidence regarding the chemical composition, the functional activities, and the related mechanisms. Secondly, few of the existing studies have correlated the chemical composition of this species with its biological effects. Moreover, existing studies have revealed different aspects related to the chemical composition of the species, but their functional value has not been tested. Studies that have both realized the evaluation of chemical composition and correlated them with different functional activities are not exhaustive and have left numerous paths open for further research.

## 5. Conclusions and Future Perspectives

The consumption of wild mushroom species is increasingly reported worldwide, and thus, updated knowledge regarding nutritional and pharmacological profiles is required. The present review has examined the valuable and underexploited nutraceutical potential of the edible mushroom *C. cornucopioides*. The present study emphasized both the distinct attributes of this mushroom related to the nutritional value supported by macro- and micronutrients (glucides, proteins, amino acids, fatty acids, vitamins, minerals) as well as the functional properties (immunomodulatory, anti-inflammatory, antimutagenic, cytotoxic, antimicrobial properties) provided by the compounds biologically active from the classes of polyphenols (flavonoids, phenolic acids), terpenoids (sesquiterpenoids, sterols), and polysaccharides. *C. cornucopioides* is a very important dietary source that occupies an important position among the fungal species with important medicinal and nutritional values. In order to preserve this invaluable resource and to harness, at a maximum, its potential, there is a pregnant need to support future research studies that can offer the phytochemical and pharmacological foundation of *C. cornucopioides* but, at the same time, the present study can also bolster its nutritional uses. Future research directions of study on this species may be related to exploring the effects of different extraction methods on the bioactivity of different metabolites, analyzing, in more detailed manners, the mechanisms of action that these compounds have at the basis of their functional activity, conducting clinical trials to validate the health benefits in humans, or investigating the potential environmental impacts of harvesting *Craterellus cornucopioides*. Regarding the nutritional aspects related to the species, future perspectives may be related to testing the toxicological profile of the species. As this species is a wild mushroom and it is highly known that wild mushrooms could be rich sources of minerals (related to their characteristic ability to accumulate different macro- and microelements), the presence of toxic metals As, Hg, Cd, and Pb should also be investigated for safety reasons. Not least, obtaining new extracts through the use of eco-friendly solvents and of appropriate modern extraction techniques could better value this edible mushroom in the prevention and treatment of various diseases, emphasizing its quality as a functional food.

## Figures and Tables

**Figure 1 nutrients-16-00831-f001:**
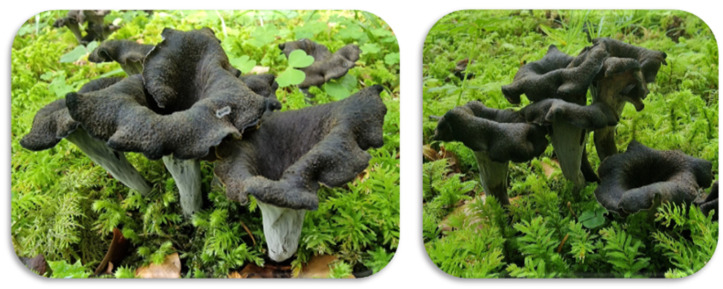
*Craterellus cornucopioides* (L.) Pers.

**Figure 2 nutrients-16-00831-f002:**
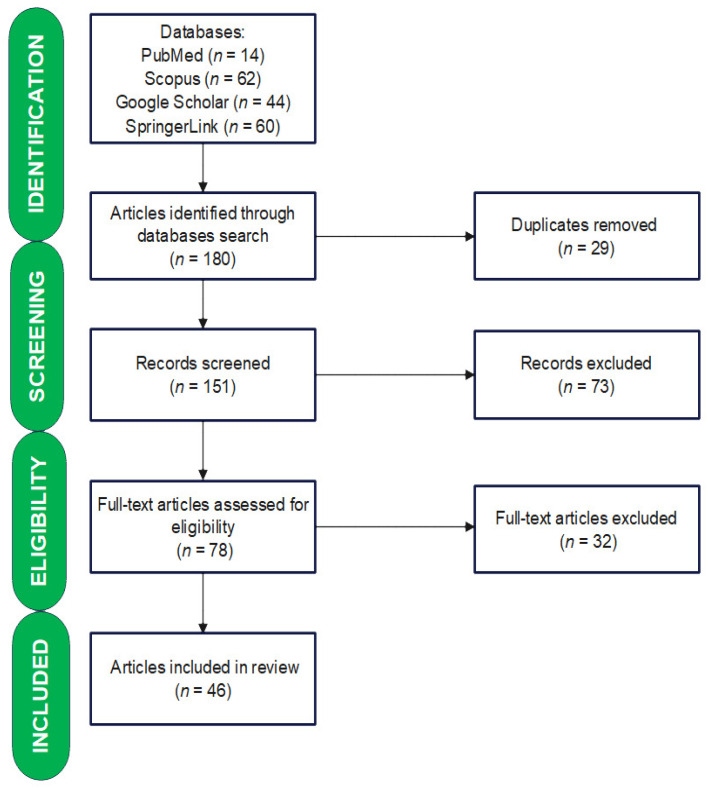
Study selection process.

**Figure 3 nutrients-16-00831-f003:**
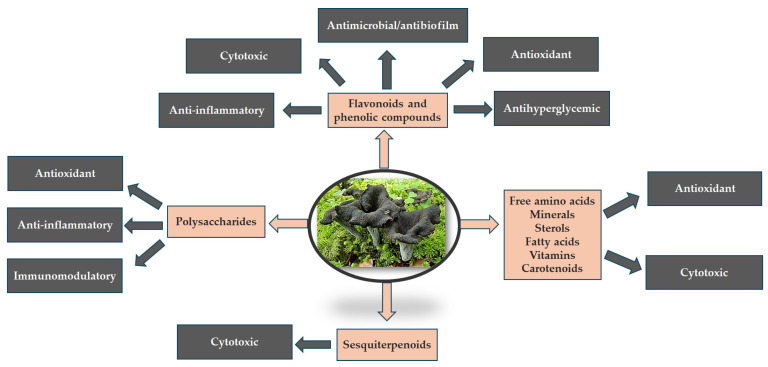
Correlation between the main classes of compounds found in the chemical composition of *C. cornucopioides* and their functional activities.

**Table 1 nutrients-16-00831-t001:** Classes of compounds and specific metabolites reported in the chemical composition of *C. cornucopioides* extracts/vegetal products.

Class of Compounds	Metabolite (s)	Extract/Vegetal Product	Reference
Flavonoids and phenolic compounds	Ferulic acid, Gallic acid, p-hydroxybenzoic acid, Protocatechuic acid, Chlorogenic acid, Caffeic acid, Syringic acid, Rosmarinic acid Pyrogallol, Catechin, Epicatechin, Epigallocatechin gallate Rutin, Quercetin, t-resveratrol, piceatannol	Methanolic extract Ethanolic extract Aqueous extract	[8,11,15,19,20,21,22,23]
Polysaccharides	CCPs-1, CCPs-2 (acidic polysaccharides with different contents of uronic acid, rhamnose, fucose, arabinose, xylose, mannose, glucose, and galactose) CCPP-1 (a water-soluble polysaccharide, a heteropolysaccharide consisting of mannose, glucose, xylose, arabinose, and fructose)	Aqueous extract after removal of pigments and lipids with ethanol	[13,14,18,24,25,26,27,28]
Sesquiterpenoids	Craterellins A–C Gymnomitr-3-en-10β,15-diol Illudin F, illudin M, illudin T Illudalenol	Cultures of the mushroom Ethyl acetate extract	[29,30]
Keto esters	4-oxohex-1,6-diyl diacetate, 4-oxohex-5-enyl acetate, 6-hydroxy-4-oxohexyl acetate	Ethyl acetate extract	[31]
Triacylglycerol	Glycerol tri-dehydrocrepenynate (octadeca-9Z, 14Z-dien-12-ynoate)	Chloroforme + methanol extract after removal of lipases with isopropanol	[32]
Water soluble vitamins	Vitamins C, B1, B2, B3 and B6 Vitamin B12	Methanolic extract Aqueous extract	[15,33]
Fat-soluble vitamins	Vitamins A, D3 and E	Cyclohexane and Dichloromethane extract	[15]
Free amino acids	Threonine, Methionine, Valine, Leucine/isoleucine, Phenylalanine (essential amino acids) Arginine, Serine, Aspartic acid, Glutamic acid, Glycine, Alanine, Tyrosine, Proline (non-essential amino acids)	Dried mushroom	[15,21]
Fatty acids	C 15:0, C 16:0, C 16:1, C 17:0, C 18:0, C 18:1n9c, C 18:2n6c, C 20:0, C 20:5n3, C 22:0, C 22:6n3, C 24:0, C 24:1n15c, SFAs, MUFAs, PUFAs	Cyclohexane extract	[15,21,34]
Sterols	Ergosterol, brassicasterol, campesterol, stigmasterol, β-sitosterol	Petroleum ether extract Chloroforme + methanol extract	[21,34,35]
Carotenoids	β-carotene and lycopene	Methanolic extract	[19]
Minerals	Fe, Zn, K, Na, Ca, Mn, Cu, Mg, Pb, Cd, As, P, Li	Fresh mushroom	[21,36,37,38,39]

**Table 3 nutrients-16-00831-t003:** The correlation between the functional activities of *C. cornucopioides* extracts and the mushroom’s chemical composition, highlighted via comparison with the most well-known edible mushrooms.

Class of Compounds	Functional Activity	Edible Mushrooms Comparatively Tested	Reference
Flavonoids and phenolic compounds Free amino acids Fatty acids Sterols Minerals	α-glucosidase-inhibitory activity	*Clitocybe maxima > C. cornucopioides > Stropharia rugoso-annulata* > *Laccaria amethystea > Catathelasma ventricosum* (ethanolic extract) *Stropharia rugoso-annulata > C. cornucopioides > Clitocybe maxima > Laccaria amethystea > Catathelasma ventricosum* (ethyl acetate extract)	[21]
	α-amylase-inhibitory activity	*C. cornucopioides > Clitocybe maxima > Laccaria amethystea > Stropharia rugoso-annulata > Catathelasma ventricosum* (ethanolic extract) *Clitocybe maxima > Stropharia rugoso-annulata > C. cornucopioides > Laccaria amethystea > Catathelasma ventricosum* (ethyl acetate extract)	
	Antioxidant—DPPH assay	*Clitocybe maxima > C. cornucopioides > Stropharia rugoso-annulata > Laccaria amethystea > Catathelasma ventricosum* (ethanolic extract) *C. cornucopioides > Stropharia rugoso-annulata > Clitocybe maxima > Laccaria amethystea > Catathelasma ventricosum* (ethyl acetate extract)	
	Antioxidant—ferrous-ion-chelating-activity assay	*Stropharia rugoso-annulata > C. cornucopioides = Catathelasma ventricosum > Laccaria amethystea > Clitocybe maxima* (ethanolic extract) *Clitocybe maxima > Laccaria amethystea > Stropharia rugoso-annulata > C. cornucopioides > Catathelasma ventricosum* (ethyl acetate extract)	
	Antioxidant—Reducing-power assay	*Clitocybe maxima > Catathelasma ventricosum > Stropharia rugoso-annulata > C. cornucopioides > Laccaria amethystea* (ethanolic extract) *Stropharia rugoso-annulata > Catathelasma ventricosum > Clitocybe maxima > C. cornucopioides > Laccaria amethystea* (ethyl acetate extract)	
Flavonoids and phenolic compounds	Antioxidant—DPPH, ABTS assay	*Pleurotus ostreatus > C. cornucopioides > Agaricus bisporus > Tuber melanosporum > Marasmius oreades* (fluidized bed ethanol extracts)	[22]
	Antioxidant—ferrous-ion-chelating-activity assay	*Tuber melanosporum > C. cornucopioides > Marasmius oreades > Agaricus bisporus > Pleurotus ostreatus* (fluidized bed ethanol extracts)	
	Antioxidant—inhibition-of-erythrocyte-hemolysis assay	*Marasmius oreades > Tuber melanosporum > Pleurotus ostreatus > C. cornucopioides > Agaricus bisporus* (fluidized bed ethanol extracts)	
	Anti-inflammatory—xanthine oxidase activity assay	*Tuber melanosporum> Pleurotus ostreatus > C. cornucopioides > Agaricus bisporus* (fluidized bed ethanol extracts)	
	Anti-inflammatory—lipoxygenase inhibition assay	*C. cornucopioides > Agaricus bisporus > Tuber melanosporum> Pleurotus ostreatus* (fluidized bed ethanol extracts)	
Flavonoids and phenolic compounds	Cytotoxic	*C. cornucopioides = Cantharellus cibarius > Cantharellus cinereus > Hydnum repandum > Calocybe gambosa* (methanolic extracts)	[23]
Flavonoids and phenolic compounds	Antioxidant—inhibition-of-linoleic-acid-oxidation assay	*Cantharellus cibarius > C. cornucopioides > Lactarius deliciosus > Calocybe gambosa > Hygrosphorus marzuolus > Boletus edulis > Pleurotus ostreatus > Agaricus bisporus* (methanolic extract)	[20]

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
