# Peer review of "Comprehensive Review of Functional and Nutraceutical Properties of Craterellus cornucopioides (L.) Pers."

_nutrients, 2024, doi:10.3390/nu16060831_

Round 1

Reviewer 1 Report

Comments and Suggestions for Authors

General Comments:

The manuscript by Mariana-Gabriela Bumbu., et al presents a thorough review of Craterellus cornucopioides in aggregating and synthesizing available data on this species of edible mushroom, offering valuable insights into its potential for nutraceutical and therapeutic applications. The review methodically categorizes the significant compounds found within Craterellus cornucopioides, such as polysaccharides, phenolic compounds, terpenes, and various others, highlighting their contributions to the mushroom's functional properties like antioxidant, antimicrobial, anticancer, and anti-inflammatory activities. However, there are areas where the review could be enhanced: 

Comments:

1, Coverage and Comparative Analysis: The review provides extensive coverage of the nutritional and pharmacological aspects of Craterellus cornucopioides, which is commendable. It would be beneficial if the review could also highlight any existing research gaps or inconsistencies within the literature, offering a more critical analysis of the current body of knowledge. While there are occasional comparisons with other mushroom species, a more detailed comparative analysis could underline Craterellus cornucopioides' unique value. This could involve more systematic tabulation or visual representation of its nutritional and pharmacological properties relative to other species.

2, Graphical Content: The addition of more figures, tables, and possibly charts summarizing the main findings, such as the chemical constituents and their corresponding bioactivities and mechanism of actions could aid readers in quickly grasping the key takeaways. Visual representations of the molecular structures of significant compounds or pathways involved in the pharmacological effects could also enhance comprehension. 

3, Future Directions: More explicit suggestions for future research directions would enhance the review. Identifying specific gaps in the current knowledge and proposing methodological approaches to address these could set a constructive path for upcoming studies. For instance, exploring the effects of different extraction methods on the bioactivity of compounds, conducting clinical trials to validate health benefits in humans, or investigating the potential environmental impacts of harvesting Craterellus cornucopioides could offer clear directions for subsequent investigations.

Author Response

Dear Editor,

On behalf of my co-authors, I would like to submit the revised form of the manuscript entitled “Comprehensive Review on Functional and Nutraceutical Properties of Craterellus cornucopioides (L.) Pers.”, sent for publication in the “Nutrients” journal, together with the response to the reviewers’ comments after the first round of review, which are found below. The necessary suggested specific changes were performed and, where necessary, supplementary information or clarification was added, to respond to the main points that are raised by the reviewers. Changes are highlighted in by highlighting them in red, for the benefit or editors and reviewers.

Reviewer 1:

General Comments:

The manuscript by Mariana-Gabriela Bumbu., et al presents a thorough review of Craterellus cornucopioides in aggregating and synthesizing available data on this species of edible mushroom, offering valuable insights into its potential for nutraceutical and therapeutic applications. The review methodically categorizes the significant compounds found within Craterellus cornucopioides, such as polysaccharides, phenolic compounds, terpenes, and various others, highlighting their contributions to the mushroom's functional properties like antioxidant, antimicrobial, anticancer, and anti-inflammatory activities. However, there are areas where the review could be enhanced: 

Authors thank the reviewer for the suggestions that helped improve the quality of the manuscript. Please find below the on-point responses to every of the suggestions found in the review form:

Comments:

1, Coverage and Comparative Analysis: The review provides extensive coverage of the nutritional and pharmacological aspects of Craterellus cornucopioides, which is commendable. It would be beneficial if the review could also highlight any existing research gaps or inconsistencies within the literature, offering a more critical analysis of the current body of knowledge. While there are occasional comparisons with other mushroom species, a more detailed comparative analysis could underline Craterellus cornucopioides' unique value. This could involve more systematic tabulation or visual representation of its nutritional and pharmacological properties relative to other species.

Authors’ response: Existing research gaps or inconsistencies within the literature were added at the end of the Discussion section, offering a more critical analysis of the current body of knowledge on the mushroom that represents the subject of the study. Moreover, in the same section, a table highlighting correlation between classes of compounds, functional activities and species that this species was compared to, was added, in this way emphasizing once again, as in the rest of the manuscript, the unique value of Craterellus cornucopioides. Some additional comments were also added, in order to sustain more all of these. A figure representing correlation between the main classes of compounds and the corresponding functional activities was also added in this section, for the same reason, to emphasize the unique value of this fungal species.

2, Graphical Content: The addition of more figures, tables, and possibly charts summarizing the main findings, such as the chemical constituents and their corresponding bioactivities and mechanism of actions could aid readers in quickly grasping the key takeaways. Visual representations of the molecular structures of significant compounds or pathways involved in the pharmacological effects could also enhance comprehension. 

Authors’ response: The manuscript contains a total of 3 tables: the first sumarizing the chemical composition of the species, with the classes of compounds and their representatives, the second sumarizing the functional activities of the species and their corresponding mechanisms and the third one (recently added, at the suggestion of the reviewer 1), correlating the main classes of compounds with their corresponding functional activity and species that they were studied in comparison with. To enhance the comprehension of main ideas in the present manuscript, a figure making correlation between the main classes of compounds and their corresponding functional activities was added. In correlation with Table 2, containing pathways involved in the pharmacological effects, authors hope to help highlighting key takeaways.

3, Future Directions: More explicit suggestions for future research directions would enhance the review. Identifying specific gaps in the current knowledge and proposing methodological approaches to address these could set a constructive path for upcoming studies. For instance, exploring the effects of different extraction methods on the bioactivity of compounds, conducting clinical trials to validate health benefits in humans, or investigating the potential environmental impacts of harvesting Craterellus cornucopioides could offer clear directions for subsequent investigations.

Authors’ response: The authors thank the reviewer for the constructive examples regarding the future perspectives of the studies and totally agree with them. All of them can be found in the last section of the manuscript, together with other possible paths to follow.

On behalf of the authors of the manuscript, I thank you for the consideration to our work and the reviewers for the suggestions that helped to significantly improve the quality of the manuscript.

Sincerely yours,

Lecturer Irina Ielciu, PhD

Reviewer 2 Report

Comments and Suggestions for Authors

The present manuscript aimed to review the current data regarding the morphology, chemical profile and medicinal potential of the black trumpet mushroom. The authors conducted a search on Four databases (PubMed, Scopus, Google Scholar and SpringerLink). In total, 180 references were found, of which 14 on PubMed, 62 on Scopus, 44 on Google Scholar and 60 on SpringerLink. After removing duplicates (29), article titles and abstracts were manually screened to exclude studies not related to the topic.

The following inclusion criteria were used for seclection: studies published in English, full-text availability, the presence of the keyword Craterellus cornucopioides in the title and/or abstract. The exclusion criteria were as follows: the study topic (the absence of any information regarding the chemical composition or biological effects), conference papers, records with no full text available. At last, 46 articles were included in the review.

In this way, the authors described the chemical composition, Immunomodulatory, anti-inflammatory, antimutagenic, antioxidant, cytotoxic, antimicrobial properties of black trumpet mushroom.

 My only suggestion is that the authors correlate the biological properties with the chemical composition. Which compounds (described in the chemical composition section) are responsible for the biological properties described? A description and a table summarizing this information would be interesting.

Author Response

Dear Editor,

On behalf of my co-authors, I would like to submit the revised form of the manuscript entitled “Comprehensive Review on Functional and Nutraceutical Properties of Craterellus cornucopioides (L.) Pers.”, sent for publication in the “Nutrients” journal, together with the response to the reviewers’ comments after the first round of review, which are found below. The necessary suggested specific changes were performed and, where necessary, supplementary information or clarification was added, to respond to the main points that are raised by the reviewers. Changes are highlighted in by highlighting them in red, for the benefit or editors and reviewers.

Reviewer 2:

The present manuscript aimed to review the current data regarding the morphology, chemical profile and medicinal potential of the black trumpet mushroom. The authors conducted a search on Four databases (PubMed, Scopus, Google Scholar and SpringerLink). In total, 180 references were found, of which 14 on PubMed, 62 on Scopus, 44 on Google Scholar and 60 on SpringerLink. After removing duplicates (29), article titles and abstracts were manually screened to exclude studies not related to the topic.

The following inclusion criteria were used for seclection: studies published in English, full-text availability, the presence of the keyword Craterellus cornucopioides in the title and/or abstract. The exclusion criteria were as follows: the study topic (the absence of any information regarding the chemical composition or biological effects), conference papers, records with no full text available. At last, 46 articles were included in the review.

In this way, the authors described the chemical composition, Immunomodulatory, anti-inflammatory, antimutagenic, antioxidant, cytotoxic, antimicrobial properties of black trumpet mushroom.

Authors thank the reviewer for the suggestions that helped improve the quality of the manuscript. Please find below the on-point responses to every of the suggestions found in the review form:

My only suggestion is that the authors correlate the biological properties with the chemical composition. Which compounds (described in the chemical composition section) are responsible for the biological properties described? A description and a table summarizing this information would be interesting.

Authors’ response: In the Discussion section, a figure containing correlations between the main classes of compounds and their corresponding biological activities was added. Moreover, in the same section a table presenting correlations between the main classes of compounds, functional activities and main edible species that were tested in comparison with the mushroom that is the subject of this work, was added. In this way, authors hope to enhance compresension of the main key findings of the present work.

On behalf of the authors of the manuscript, I thank you for the consideration to our work and the reviewers for the suggestions that helped to significantly improve the quality of the manuscript.

Sincerely yours,

Lecturer Irina Ielciu, PhD

Round 2

Reviewer 1 Report

Comments and Suggestions for Authors

The authors have carefully and thoughtfully addressed the concerns raised in previous review circle. 

The manuscript has been improved by adding more information and discussion.  The Table 3 particularly provides informative knowledge by comparing among different edible mushroom species. 

Thanks!